# Greater Horseshoe Bats Recognize the Sex and Individual Identity of Conspecifics from Their Echolocation Calls

**DOI:** 10.3390/ani12243490

**Published:** 2022-12-10

**Authors:** Xiao Tan, Aiqing Lin, Keping Sun, Longru Jin, Jiang Feng

**Affiliations:** 1Jilin Provincial Key Laboratory of Animal Resource Conservation and Utilization, Northeast Normal University, Changchun 130000, China; 2Jilin Provincial Engineering Laboratory of Avian Ecology and Conservation Genetics, Northeast Normal University, Changchun 130000, China; 3College of Life Science, Jilin Agricultural University, Changchun 130000, China

**Keywords:** bats, echolocation calls, acoustic recognition, communication, playback experiment

## Abstract

**Simple Summary:**

Recognition is crucial for many aspects of an animal’s life and directly affects its fitness. The aim of this study was to investigate whether greater horseshoe bats can recognize a conspecific’s sex and individual identity from its echolocation call alone. Through playback experiments, we demonstrated that greater horseshoe bats can recognize the sex and individual identity of conspecifics from their echolocation calls, which indicates that the echolocation calls of greater horseshoe bats may have potential communication functions. The results of this study improve our understanding of the communication function of the echolocation calls of bats.

**Abstract:**

The echolocation calls of bats are mainly used for navigation and foraging; however, they may also contain social information about the emitter and facilitate social interactions. In this study, we recorded the echolocation calls of greater horseshoe bats (*Rhinolophus ferrumequinum*) and analyzed the acoustic parameter differences between the sexes and among individuals. Then, we performed habituation-discrimination playback experiments to test whether greater horseshoe bats could recognize the sex and individual identity of conspecifics from their echolocation calls. The results showed that there were significant differences in the echolocation call parameters between sexes and among individuals. When we switched playback files from a habituated stimuli to a dishabituated stimuli, the tested bats exhibited obvious behavioral responses, including nodding, ear or body movement, and echolocation emission. The results showed that *R. ferrumequinum* can recognize the sex and individual identity of conspecifics from their echolocation calls alone, which indicates that the echolocation calls of *R. ferrumequinum* may have potential communication functions. The results of this study improve our understanding of the communication function of the echolocation calls of bats.

## 1. Introduction

Recognition refers to the process of identifying and distinguishing individual animals through physiological characteristics (e.g., voice, odor, and facial pattern) and behavioral characteristics [1]. For animals with complex ecological communities, their social behaviors depend on recognition because recognition is usually the basic premise of the occurrence of social behaviors and plays an important role in their survival and reproduction [2,3]. Recognition ranges across a wide spectrum, including kin, mates, sex, neighbors, rivals, species, predators, and prey [4]. Recognition is crucial for many aspects of an animal’s life and directly affects its fitness [3], such as forming interspecific foraging associations to gain protection against predators and/or to enhance feeding efficiency [5], and eavesdropping on another individual’s calls to obtain information about foraging sites [6] or habitats [7].

Recognition can be mediated through various modes of communication including vision, olfaction, tactile, spatial memory, auditory, and combinations of these [8]. Since acoustic signals have their own advantages, such as the ability to be perceived over large distances and being only slightly limited by environmental barriers, they are usually used as an important recognition cue for vocal animals, such as amphibians, birds, and mammals. For example, the olive frog (*Babina adenopleura*) can recognize conspecific neighbors and strangers based on their individual vocal signatures, and they display lower aggression toward the neighbors than toward strangers [9]. Owls (*Otus insularis*) can determine whether the caller is a companion or prey through acoustic signals, and then, they decide to communicate with or prey upon the caller [10]. Female seals (*Phoca vitulina*) can recognize the calls of their offspring to achieve mother–infant reunions [11,12].

Bats (Chiroptera) are the only mammals capable of flight. The majority of insectivorous bats are highly gregarious and often form large colonies. They usually rely little upon their vision due to their nocturnal lifestyles, and they mainly use echolocation calls to navigate and forage in dark environments [13]. However, a growing number of studies have shown that the echolocation calls of bats can also encode information about the caller and facilitate their social interactions [14,15,16,17]. For example, several studies have shown that bats can recognize the following from echolocation calls: the identity of the species [18,19], group membership [20,21], family affiliation [22], age [23,24], sex [25], and individual identity [26]. Recently, the results of habituation-dishabituation playback experiments have shown that *Rhinolophus clivosus* discriminate between sex and individuals via their echolocation pulses alone, and researchers have found that the echolocation calls of *R. clivosus* have significant structural variations that may be helpful for reliable individual identification in large groups comprised of many individuals [27]. These studies reported that echolocation calls have a communicative function and play an important role in intraspecific social interactions [13,28,29]. However, while research on the acoustic communication of bats is rapidly growing, it still lags behind other mammalian taxa in general compared to the high species diversity of bats (over 1400 species worldwide [30]) [13].

The greater horseshoe bat, *Rhinolophus ferrumequinum,* is widely distributed in Asia and Europe [31]. They are usually gregarious and mainly rely on echolocation calls for spatial navigation and foraging. Their echolocation calls predominantly consist of a long constant frequency (CF) component, which is often initiated and terminated by brief frequency-modulated (FM) sweeps of a substantial bandwidth [32]. Möhres suggested that *R. ferrumequinum* may be able to distinguish between individuals from listening to their echolocation pulses [33]. A recent playback study provided experimental evidence that greater horseshoe bats can distinguish acoustically similar heterospecific species based on their echolocation calls [19]. Another playback study demonstrated that greater horseshoe bats can discriminate between the echolocation calls of its local population and a foreign one, which may promote their assortative mating and reproductive isolation [34]. However, to the best of our knowledge, thus far, no study has tested whether greater horseshoe bats can recognize the sex and individual identity of an unfamiliar conspecific from its echolocation call through playback experiments. Therefore, the aim of this study was to investigate whether greater horseshoe bats can recognize a conspecific’s sex and individual identity from its echolocation call alone. Based on the increasing evidence that echolocation calls contain social information, we predicted that (1) the echolocation calls of *R. ferrumequinum* have sexual and individual characteristics with statistically significant differences; and (2) the bats are able to recognize the sex and individual identity of a caller by its echolocation call.

## 2. Materials and Methods

### 2.1. Collection and Husbandry of Bats

In May and July 2016, *Rhinolophus ferrumequinum* were captured using mist nets after sunset in Dalazi Cave (125.51° E, 41.03° N), Ji’an City, and Xin Cave (125.89° E, 43.28° N), Panshi City, in Jilin Province, China. Their sex and reproductive status were determined based on their external morphological characteristics [35]. Pregnant females, lactating females, and nursing pups were not collected to minimize the disturbance of maternity colonies. Thus, eight adult greater horseshoe bats (four males and four females) were collected in Dalazi Cave, and 20 adult greater horseshoe bats (10 males and 10 females) were collected in Xin Cave. A numbered alloy band (4.2 mm internal width, 5.5 mm height, Porzana Ltd., Winchelsea, UK) was placed on the forearm of each bat for individual identification. These manipulations had no negative effects on the bats’ behavior or health based on our prior studies [36,37].

The bats captured from the two collection sites were transferred to and housed separately in two rooms of the same size (4 m long × 3 m wide × 3 m high) to avoid familiarization between the groups via sound signals. The bats were able to fly freely in the husbandry room. For the first week, the bats were trained to actively look for food (larvae of yellow mealworms, *Tenebrio molitor*) and water until all of the bats could feed themselves. Vitamins and calcium tablets were ground into powder as supplements and added to the feeding box of yellow mealworms so that they could feed on these supplements while eating bran, thus indirectly supplementing the bats with nutrients. In the laboratory, the ambient temperature (26 ± 2 °C), relative humidity (55 ± 5%), and light (lights on: 06:00–18:00; lights off: 18:00–06:00) were set to simulate the environmental parameters of the natural habitat of *R. ferrumequinum*. The husbandry rooms were decorated with numerous artificial plants, which were randomly hung from the ceiling and attached to the walls, to enrich the environment.

### 2.2. Sound Recording and Editing

#### 2.2.1. Sound Recording

To analyze the acoustic characteristics and create the playback files, we recorded the echolocation calls of the individual bats between 1900 and 2100 h when the bats were alert in a sound recording room (5 m long × 2 m wide × 2.7 m high). A single bat was perched in an experimental cage (60 × 60 × 60 cm) surrounded by sound-attenuating foam during the acoustic recording. An ultrasonic microphone (UltraSoundGate CM16/CMPA, Avisoft Bioacoustics, Berlin, Germany) was connected to an ultrasound recording interface with a sampling rate of 250 kHz and a 16-bit resolution, and every 60 s, the wave file generated was stored in a battery-powered notebook computer. The echolocation calls of each individual were recorded at a distance of 1 m from in front of the focal bat head. To ensure that the subsequent analysis and editing requirements were met, at least five 60 s sound files were recorded for each individual.

#### 2.2.2. Playback Stimuli for Sex Recognition

The recorded echolocation calls of the eight adult bats from Dalazi Cave were edited using Avisoft-SASLab Pro (Version 5.1) to form a single-sex playback file consisting of four same-sex individuals. For each bat, the call series were divided into shorter segments with durations ranging from 100 to 1000 ms, within which high signal-to-noise ratio (>30 dB) segments were then randomly selected and combined with segments from other individuals to create playback files. Each file contained recordings of four bats of the same sex, and each bats’ recording contained at least 110 echolocation pulses, with a playback file length of 35 s, i.e., 15 s of echolocation calls +5 s of silent interval +15 s of echolocation calls (Appendix A). We edited five playback files for each sex, i.e., five female playback files and five male playback files. The echolocation call spectrograms of the different sexes of *R. ferrumequinum* are shown in Figure 1.

#### 2.2.3. Playback Stimuli for Individual Recognition

The recorded echolocation calls of the eight bats from the Dalazi Cave and the 20 bats from Xin Cave were edited using Avisoft-SASLab Pro Version 5.1 (Avisoft Bioacoustics, Berlin, Germany) to form playback files for the different individuals. Each bat call series was divided into shorter segments of 100–1000 ms in duration, and then, a randomly selected segment with a high signal-to-noise ratio (>30 dB) was edited into a playback file with a duration of 35 s, i.e., 15 s of echolocation calls +5 s of silent interval +15 s of echolocation calls (Appendix A). We did not standardize the length of the intervals between echolocation calls and retained the recorded natural call rate [34]. Five different playback files were edited for each individual for subsequent acoustic analysis (to confirm whether the calls had individual characteristics) and playback. The echolocation call spectrograms of different *R. ferrumequinum* individuals are shown in Figure 2.

### 2.3. Sound Analysis and Playback

In the two sex playback files for the Dalazi Cave greater horseshoe bats, 160 pulses with high signal-to-noise ratios (80 pulses for each sex) were randomly selected for sound analysis. We analyzed the echolocation pulses using Avisoft-SASLab Pro version 5.1 (Avisoft Bioacoustics, Berlin, Germany). Prior to conducting the acoustic analysis, the signals were normalized to 0.75 V to evaluate the quality of the waveforms and exclude overloaded signals [38]. Measurements were taken of spectrograms generated using a 1024-point fast Fourier transformation and a Hamming window with a frame size of 75% and 75% overlap.

When the echolocation calls were analyzed, the background noise was erased in Avisoft software. To quantify the sex characteristics of the echolocation calls, the call duration (s) of the harmonics at the energy maximum was measured using the automatic parameter measurements tool in Avisoft with full visual control of the spectrogram, as well as four spectrum-based parameters (kHz): the peak frequency, minimum frequency of bandwidth, maximum frequency of bandwidth, and bandwidth, for which the measurement threshold was 10 dB. For these four spectrum-based parameters, three locations (i.e., the center, end, and maximum amplitude of the element) were measured (Figure 3) [39,40,41]. Thus, a total of 13 acoustic variables were extracted: the call duration, peak frequency (end), minimum frequency of bandwidth (end), maximum frequency of bandwidth (end), bandwidth (end), peak frequency (center), minimum frequency of bandwidth (center), maximum frequency of bandwidth (center), bandwidth (center), peak frequency (maximum), maximum frequency of bandwidth (maximum), minimum frequency of bandwidth (maximum), and bandwidth (maximum). For the analysis of the individual characteristics of the echolocation calls, the playback files of 10 experimental individuals were selected from the recorded playback files, containing 16 high signal-to-noise ratio pulses each (10 individuals, 160 pulses in total). The above 13 acoustic parameters were measured and analyzed to determine the individual characteristics of their acoustic parameters.

Playback experiments using the habituation-dishabituation (also known as habituation-discrimination) method were conducted in an experimental iron wire mesh cage (0.45 m long × 0.35 m wide × 0.35 m high; Figure 4) between 19:00 and 21:00 when the bats were alert. One ultrasonic loudspeaker (ScanSpeak, Avisoft Bioacoustics, Berlin, Germany), one infrared camera (HDR-CX760E, Sony Corp., Tokyo, Japan), and one ultrasonic microphone (UltraSoundGate CM16/CMPA, Avisoft Bioacoustics, Berlin, Germany) were fixed to three tripods. The speaker was connected to an ultrasound recording interface (Avisoft UltraSoundGate116H, Avisoft Bioacoustics, Berlin, Germany), and the microphone was connected to an ultrasound recording system (UltraSoundGate 116). The speaker and microphone were placed 0.60 m from the shorter wall of the cage and were directed toward the tested bat.

In the sex recognition playback experiments, the subject bat from Xin Cave was placed in the experimental cage. One type of playback file (5 female files or 5 male files) was selected randomly and played back as the habituation file with a sound pressure level amplitude of around 60 dB at the site where the bat was hanging. We defined habituation as the bat becoming motionless, with no body or head movements, no crawling activity, no stretching of wings and legs, and no echolocation calls for 30 s [34]. After the bat remained habituated for 30 s, we switched the habituation file to the dishabituation file. The four types of playback combinations for the sex recognition playback experiment were MM, MF, FM, and FF (M for male, F for female). If the bat showed any behavioral activity (nodding, ear movement, body movement, or echolocation emission), we took this as evidence that the bat had discriminated between the two playback signals [34]. If a bat remained habituated, we considered this as indication that the bat regarded the new signal as belonging to the same class as the habituation signal [19,42]. After the bat was dishabituated, i.e., completely stationary, including no nodding, no ear and body movement, and no echolocation emission, we immediately switched to a control stimulus, i.e., 500 ms of white noise, to check whether the bat was listening to the playback sound during the entire playback period and was not asleep. If the bat restarted its activity, it proved that it had not fallen asleep. If the bat did not show any behavioral response, the bat was replaced with another and the experiment was repeated, and the data for the first bat were excluded from the statistical analysis.

In the individual recognition playback experiments, first, the individual’s own playback file was played as the habituated file. After the bat was habituated to this sound and remained motionless for 30 s as described above, the playback files of different individuals were played. The playback combinations were AA’ and AB, where A and B represented two different individuals; AA’ represented the transfer of the habituated sound file from individual A to another sound file of individual A, and AB represented the transfer of the habituated sound file from individual A to the dishabituated sound file of individual B. Each experimental individual was subjected to only one combination of playback experiments per day, and after each type of playback experiment was completed for the 20 individuals, the next type of playback experiment was performed one week later.

### 2.4. Behavioral Observations and Response Measures

The videos recorded during the playback experiments were observed to measure the behavioral responses of the bats to the echolocation calls of the different sexes and individuals after converting the playback files in each set of playback experiments. All of the observed behaviors were counted according to the following criteria.

(1)Nodding. This behavior was characterized by a movement of the head to the chin. The number was counted as 1 each time the subject bat raised its head and then lowered its head. If the bat only raised its head, the number was counted as 0.5.(2)Ear movement: movement of the ears around to detect the caller. This was often accompanied by echolocation. The number was counted as 1 when the bat moved its left or right ear.(3)Body movement: rotation of the body toward the microphone to aid in detection. The number was counted as 1 when the bat turned to the left or right.(4)Echolocation calls. The number of echolocation pulses emitted by the bat was counted.

To minimize observer bias, a blind method was used so that the observer who measured the bats’ behavioral responses had no knowledge of their treatment or identity.

### 2.5. Statistical Analysis

To reduce the multicollinearity between the acoustic variables of the echolocation calls, all of the measured parameters were analyzed using principal component analysis (PCA) in SPSS 22.0 (IBM Corp., Armonk, NY, USA). For the playback files of the echolocation calls for sex recognition, five principal components (eigenvalues > 1) were extracted from the dimensionality reduction of the 13 parameters, which explained 85.0% of the total variance. Of these five components, the minimum frequency of bandwidth (maximum) (0.92), peak frequency (maximum) (0.91), maximum frequency of bandwidth (maximum) (0.88), and maximum frequency of bandwidth (end) (0.82) contributed the most. For the playback files of the echolocation calls for individual recognition, four principal components (eigenvalues > 1) were extracted, explaining 84.3% of the total variance. The maximum frequency of bandwidth (maximum) (0.95), minimum frequency of bandwidth (maximum) (0.91), peak frequency (maximum) (0.90), and bandwidth (end) (0.86) contributed the most.

SYSTAT13.0 (Systat Software, Inc., San Jose, CA, USA) was used to present the principal component data of the acoustic parameters of the sex recognition experiment on confidence ellipses. A multivariate analysis of variance (MANOVA) was performed on the echolocation calls in the sex recognition experiment to compare the differences in the acoustic parameters between the calls of the different sexes. Permuted discriminant function analysis (pDFA) [43] was performed on the principal components extracted from the individual recognition echolocation acoustic parameters to compare the significance of the classification success with that of the random classification and to determine their individual characteristics. Pearson chi-square tests were conducted to determine whether there was a significant difference in the proportion of bats that showed any behavioral response after switching the acoustic stimuli between different types of playback combinations. If significant differences were found, pairwise comparisons were made using Fisher’s exact test. In addition, the Kolmogorov–Smirnov test was conducted to determine if the behavioral data exhibited a normal distribution. If the data were normally distributed, the t-test was used to compare the differences in the numbers of the nodding, ear movements, body movements, and echolocation pulses across the different types of test combinations; and if the data were not normally distributed, the Kruskal–Wallis H-test was applied.

### 2.6. Ethical Note

The bat capture methods and experiments used in this study conformed to the Northeast Normal University guidelines for animal behavior research. The ethical approval committee/board that approved this study is the Wildlife Conservation Office of the Jilin Forestry Department, Changchun, China. The approval number is NENU-W-2014-101. We routinely checked the animals’ health status by weighing them and observing their vigorous behavior. No injury and no death occurred during the feeding and experimental period. All of the bats were released at the site of capture when all of the trials were completed.

## 3. Results

### 3.1. Sexual Dimorphism in Echolocation Calls and Sex Recognition

The distribution of the echolocation calls of the different sexes in the two-dimensional space of the first and second principal components is shown in Figure 5. The two types were separated axially along the two principal components, but there were still some overlapping parts. A multivariable analysis of variance (MANOVA) of 160 pulses from two playback files for sex recognition revealed that there were significant differences in the acoustic parameters of the female and male echolocation calls (MANOVA: Wilk’s lambda = 0.248, *df* = 13, 146, *p* < 0.0001). Among them, the differences in the call duration, peak frequency (end), minimum frequency of bandwidth (end), maximum frequency of bandwidth (end), peak frequency (center), minimum frequency of bandwidth (center), maximum frequency of bandwidth (center), bandwidth (center), peak frequency (maximum), minimum frequency of bandwidth (maximum), and maximum frequency of bandwidth (maximum) were significant. Although there were areas of frequency overlap, the average values of the female echolocation calls had significantly higher parameter values than those of the male echolocation calls (see Table 1).

When the habituation file was F and the dishabituation file was M, four of the 20 *R. ferrumequinum* restarted nodding, ear movement, body movements, and the emission of echolocation calls. When the habituation file was F and the dishabituation file was M, 18 of the 20 *R. ferrumequinum* restarted nodding, ear movement, body movements, and the emission of echolocation calls (Figure 6a). When the habituation file was M and the dishabituation file was F, 18 of the 20 *R. ferrumequinum* restarted nodding, ear movement, body movements, and the emission of echolocation calls. When the habituation file was M and the dishabituation file was M, seven of the 20 *R. ferrumequinum* restarted nodding, ear movement, body movements, and the emission of echolocation calls (Figure 6b). The proportions of responding bats were significantly different for the four different combinations of playback types (Pearson chi-square test: *λ* = 33.17, *df* = 3, *p* < 0.01; Fisher’s exact test: *p* < 0.05). The numbers of nodding, ear movements, body movements, and echolocation pulses after conversion to different sex calls, were significantly different for *R. ferrumequinum* compared to a conversion to same sex acoustic stimuli (nodding: *p*_F_ = 0.004, *P*_M_ = 0.013; ear movements: *p*_F_ < 0.001, *P*_M_ = 0.019; body movements: *p*_F_ = 0.005, *p*_M_ = 0.016; echolocation calls: *p*_F_ < 0.001, *p*_M_ = 0.017; Figure 6a,b).

### 3.2. Individual Signature in Echolocation Calls and Individual Identity Recognition

The acoustic parameters of the echolocation calls used in the individual recognition experiment are described in Table 2. The pDFA of 160 calls from 10 individuals revealed that 44.7% of the calls were correctly classified as corresponding to individuals, which is significantly higher than the random classification probability (24.0%, *p* < 0.01).

When the habituation file was A and the stimulus file was A’, three of the 20 *R. ferrumequinum* restarted nodding, ear movement, body movements, and the emission of echolocation calls. When the habituation file was A and the dishabituation stimulus was B, 19 of the 20 *R. ferrumequinum* restarted nodding, ear movement, body movements, and the emission of echolocation calls. The proportion of bat responses was significantly different for the two types of playback combinations (Pearson chi-square test: *λ* = 28.97, *df* = 1, *p* < 0.01). The numbers of nodding, ear movements, body movements, and echolocation calls after transforming the different individual acoustic stimuli, were significantly different in *R. ferrumequinum* compared to transforming different acoustic stimuli for the same individual (all *p* < 0.05; Figure 7).

## 4. Discussion

### 4.1. Sexual Dimorphism of and Sex Identification from Acoustic Signals

Our results revealed that multiple parameters of the echolocation pulses of the greater horseshoe bats exhibited sexual dimorphism. For example, the echolocation calls of the female bats had higher peak frequencies than those of the males, which is consist with the results of other studies [44,45,46]. Therefore, the peak frequency is likely to be the main factor used for sex identification in this bat species. In contrast, in other studies, an absence of sexual dimorphism in echolocation calls was reported [47,48]. For those bat species, whether they can discriminate the sex of the conspecifics of a caller from their echolocation calls requires further investigation.

Several studies on birds (e.g., *Alca torda* [8] and *Calonectris diomedea* [49]) have also shown that their acoustic signals can contain sex information [50], and thus, other individuals can recognize the sex of the conspecifics by their calls. For example, female Steere’s liocichla (*Liocichla steerii*) can recognize male and female songs and respond more strongly to their partner’s song [51]. In this study, we compared the opposite sex acoustic stimuli to the same sex acoustic stimuli, and the majority of the subject bats responded more strongly to the opposite sex acoustic stimuli, indicating that the greater horseshoe bat can distinguish the sex based on their echolocation calls alone.

Recently, an increasing amount of evidence has shown that some bat species can recognize the sex of conspecifics (e.g., *Rhinolophus clivosus* [27] and *Eptesicus fuscus* [52,53]). In the wild, Knörnschild et al. provided experimental evidence that free-living territorial male *Saccopteryx bilineata* are capable of discriminating between conspecifics’ sexes based on their echolocation calls, which may facilitate the defense of roost territories [54]. In addition, the echolocation pulses of bats may play an important role in the choice of a mate. For example, female *Rhinolophus mehelyi* preferentially select males with high frequency calls during the mating season, and high frequency males sire more offspring [55]. In addition to mate selection, there are other benefits of the ability of bats to identify sex from conspecifics’ echolocation calls; for example, females can locate and follow other females when returning to roosts, which may reduce predation risk [56].

### 4.2. Individual Signature and Individual Recognition

Consistent with several previous studies [57,58,59], our results showed that the echolocation calls of greater horseshoe bats contain individual characteristics that allow for statistical discrimination although the correct discrimination rate in the discriminant function analysis (DFA) was low (44.7%). Low DFA classification rates for individual calls have also been reported in other mammals [60,61]. A low DFA classification rate does not imply a lack of individual recognition because the DFA results are influenced by many factors, including the similarity of the acoustic variables, the number of variables included, and the number of groups to be discriminated [62]. For example, when more individuals of *R. clivosus* were included, the classification success decreased because of the increasing overlap of the pulses of the individuals [27]. In addition, many mammals encode their acoustic individual characteristics with source characteristics, such as frequency, amplitude contours, and harmonic structure [60,63,64]. Therefore, the 13 acoustic parameters that we measured in this study may not truly represent the full range of acoustic differences among individuals.

In our habituation-dishabituation playbacks, almost all of the subject bats exhibited significant behavioral responses when the playback file was switched from a habituation stimulus to a dishabituation stimulus, which indicates that they could distinguish conspecifics’ individual identity based on echolocation calls. Recently, individual-specific signatures in echolocation pulses have been documented for several bat species, including *Myotis lucifugus* [14], *Tadarida teniotis* [15], and *Myotis Myotis* [57]. Such individual signatures of echolocation pulses can reduce the potential of a bat being confused or jammed by echoes from the pulses of a nearby conspecific [56]. Moreover, reliable individual recognition based on echolocation calls would be beneficial for social interactions, such as reciprocal interactions and cooperation [24].

## 5. Conclusions

In summary, through statistical analysis and playback experiments, we demonstrated that the echolocation calls of *R. ferrumequinum* contain sexual and individual characteristics, and these bats can recognize the sex and individual identity of other individuals based on their echolocation calls alone. The results of this study provide behavioral evidence for the potential communication function of echolocation calls, providing a deeper and more comprehensive understanding of the function of the echolocation calls of these bats.

## Figures and Tables

**Figure 1 animals-12-03490-f001:**
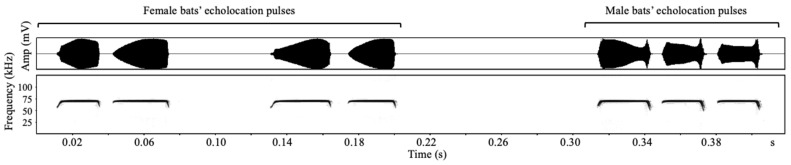
Spectrograms of a set of echolocation pulses for female (**left**) and male (**right**) *Rhinolophus ferrumequinum.*

**Figure 2 animals-12-03490-f002:**
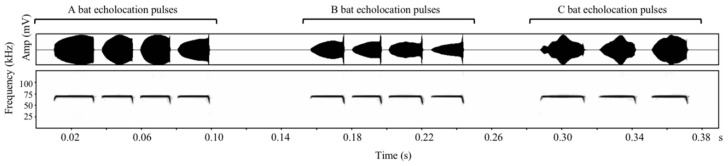
Spectrograms of a set of echolocation pulses from three different *Rhinolophus ferrumequinum* individuals (**A**–**C**).

**Figure 3 animals-12-03490-f003:**
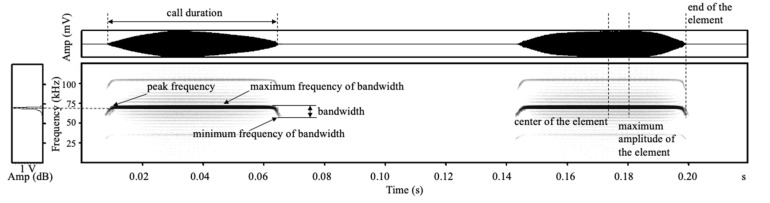
Acoustic parameters and locations of echolocation call measurements for *Rhinolophus ferrumequinum*.

**Figure 4 animals-12-03490-f004:**
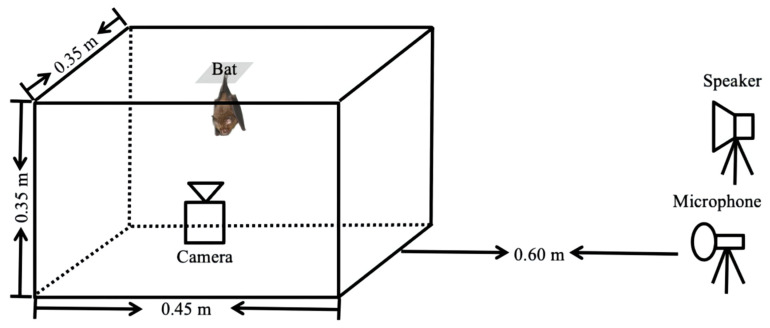
Experimental set-up of the habituation-dishabituation experiments.

**Figure 5 animals-12-03490-f005:**
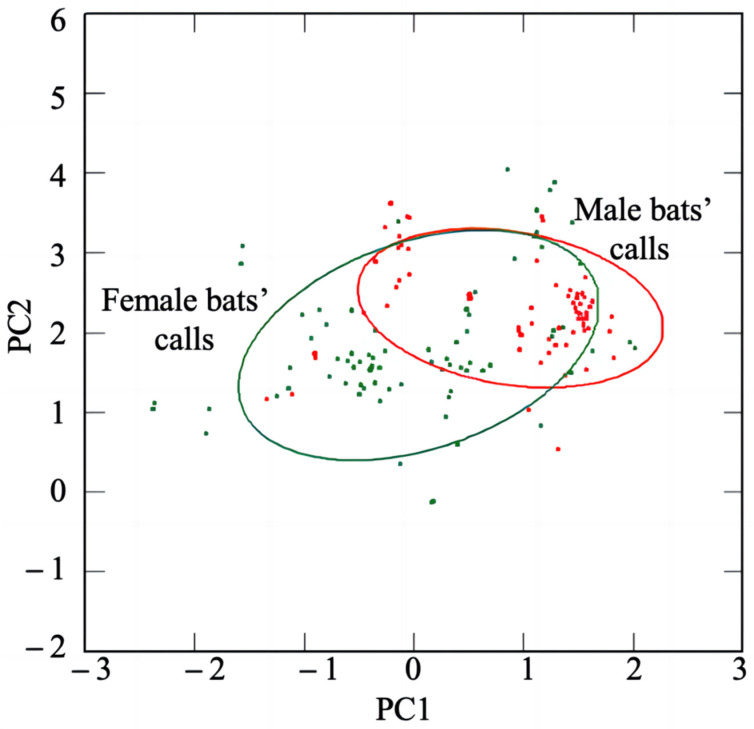
Confidence ellipse plot based on the first and second principal components. The confidence interval is 75%. The green circles denote the female calls, and the red circles denote the male calls.

**Figure 6 animals-12-03490-f006:**
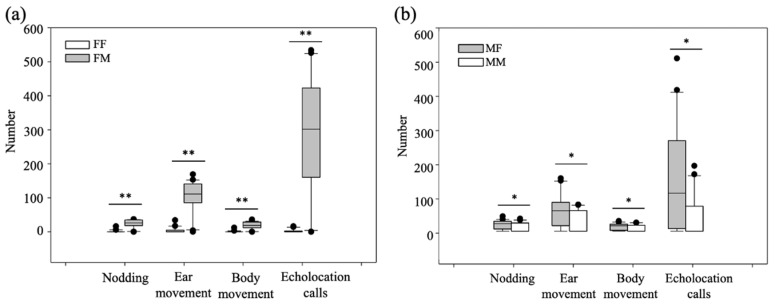
Results of the echolocation call sex recognition habituation-dishabituation playback experiment for *Rhinolophus ferrumequinum*. (**a**) The habituation sound wave is female and (**b**) the habituation sound wave is male. F represents female playback files, M represents male playback files, FF represents a change from female habituation sound waves to female dishabituation sound waves; FM represents a change from female habituation sound waves to male dishabituation sound waves; MF represents a change from male habituation sound waves to female dishabituation sound waves; and MM represents a change from male habituation sound waves to male dishabituation sound waves. * denotes *p* < 0.05, and ** denotes *p* < 0.01.

**Figure 7 animals-12-03490-f007:**
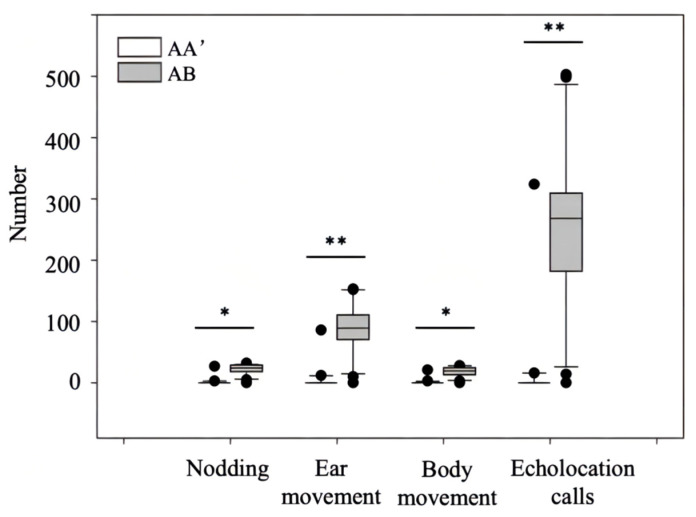
Results of the echolocation call individual identification habituation-dishabituation playback experiment for *Rhinolophus ferrumequinum*. A and B represent two different individuals. AA’ represents a change from the habituated sound file of individual A to another sound file of individual A. AB represents a change from the habituated sound file of individual A to the dishabituated sound file of individual B. * denotes *p* < 0.05, and ** denotes *p* < 0.01.

**Table 1 animals-12-03490-t001:** Description of acoustic parameters of echolocation calls of different sexes for *Rhinolophus ferrumequinum*.

Parameters	Female (Mean ± SD)	Male (Mean ± SD)	*p*-Value
Call duration	0.029 ± 0.010	0.033 ± 0.010	0.005 **
Peak frequency (end)	63.91 ± 3.28	59.32 ± 4.02	0.000 **
Minimum frequency of bandwidth (end)	58.77 ± 3.44	55.04 ± 5.39	0.000 **
Maximum frequency of bandwidth (end)	71.07 ± 1.33	69.03 ± 7.45	0.017 *
Bandwidth (end)	12.28 ± 3.66	13.94 ± 7.63	0.081
Peak frequency (center)	69.08 ± 0.42	68.90 ± 0.49	0.017 *
Minimum frequency of bandwidth (center)	67.60 ± 0.46	67.11 ± 0.37	0.000 **
Maximum frequency of bandwidth (center)	72.67 ± 0.50	72.02 ± 0.62	0.000 **
Bandwidth (center)	4.97 ± 0.42	4.83 ± 0.45	0.043 *
Peak frequency (maximum)	68.77 ± 1.14	67.87 ± 2.97	0.012 *
Minimum frequency of bandwidth (maximum)	67.01 ± 1.94	65.82 ± 3.21	0.005 **
Maximum frequency of bandwidth (maximum)	72.53 ± 0.63	71.27 ± 2.81	0.000 **
Bandwidth (maximum)	5.44 ± 1.60	5.38 ± 0.84	0.781

* denotes *p* < 0.05, ** denotes *p* < 0.01.

**Table 2 animals-12-03490-t002:** Description of acoustic parameters of echolocation calls used in the individual recognition experiment on *Rhinolophus ferrumequinum*.

Parameters	Minimum	Maximum	Mean	SD
Call duration	0.012	0.036	0.02	0.01
Peak frequency (end)	54.6	66.4	59.82	2.08
Minimum frequency of bandwidth (end)	50.7	61.5	55.06	1.69
Maximum frequency of bandwidth (end)	62.5	71.2	67.23	2.06
Bandwidth (end)	8.0	18.5	12.11	2.22
Peak frequency (center)	67.3	69.3	68.52	0.61
Minimum frequency of bandwidth (center)	65.4	67.3	66.90	0.63
Maximum frequency of bandwidth (center)	70.3	72.2	71.70	0.67
Bandwidth (center)	4.1	5.1	4.73	0.23
Peak frequency (maximum)	60.5	69.3	67.70	2.10
Minimum frequency of bandwidth (maximum)	55.6	67.3	65.68	3.18
Maximum frequency of bandwidth (maximum)	65.6	72.2	71.28	1.36
Bandwidth (maximum)	4.1	12.6	5.53	1.98

## Data Availability

The raw datasets used and analyzed during the current study are available from the corresponding author upon reasonable request.

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
