# Peer review of "Greater Horseshoe Bats Recognize the Sex and Individual Identity of Conspecifics from Their Echolocation Calls"

_animals, 2022, doi:10.3390/ani12243490_

Round 1
Reviewer 1 Report
This paper is brilliantly presented, commendably clear, and of interest to bat researchers and bioacousticians alike. I am very happy to recommend it straight to acceptance.
The introduction is excellent and covers the necessary information with clear references and is well written. The methods and results are a model of how to write-up this type of experiment and I am content that they have followed correct procedures throughout. The discussion covers everything that it should and correctly highlights that the DFA is low but that it is clearly sufficient based on the tested variables. The figures are clear and easy to follow and I love the experimental set-up figure 1.
I’ve never had a paper that I found to be so clearly written and well-presented on first review. Congratulations.
I found only one minor typo too, which was in Line 99. There is a word missing after 8. I’m guessing this is 8 adults from Dalazi Cave?
Finally, could you please just clarify in the ethics if there was an approval process for taking the animals from the wild or if it just fell under a general licence? This wasn’t quite clear.
Well done! What a great paper!
Reviewer 2 Report
The authors analysed the acoustics and made playback experiments of echolocation calls from unfamiliar individuals to demonstrate that the echolocation calls of greater horseshoe bats may contain sexual and individual characteristics. The results of this study show that these bats can recognize the sex and, at lower extent, the individual identity of other callers based on their echolocation calls alone.
I have some concerns and suggestions for the improving the manuscript.
General comments
Similar study was not considered in Introduction. There is a preceding study on another species of horseshoe bats, which the current study repeats entirely even in the methodical details: analysis of the acoustic parameters and the habituation - dishabituation paradigm of playback experiments of echolocation calls followed by analysis of movemental behaviour to demonstrate the sexual and individual characteristics in the echolocation calls (Finger N.M., Bastian A., Jacobs D.S., 2017. To seek or speak? Dual function of an acoustic signal limits its versatility in communication. Animal Behaviour, 127:135-152). However, the current manuscript only shortly mentions this study in Discussion among other studies. As the authors copy this study in many approaches, this research paradigm must be considered in Introduction and the results compared in detail in Discussion. Moreover, the results of this and the preceding study are in many aspects.
Irrelevant citing. The authors cite a few times the papers not containing the facts they refer to in the manuscript.
Methodical aspects are not perfectly clear. Creating the sequencies for playbacks is described non-transparently and thus cannot be reproduced by other researchers.
The choice of measured acoustic parameters was made extremely unsuccessfully. The authors must substantiate the selected set of acoustic parameters.
The main problem of this manuscript: the mistakes in measuring the acoustic parameters of the echolocation pulses (especially the Fundamental frequency) using Automatic Parameter Measurements in Avisoft (see details below). The authors should provide an illustrative spectrogram of an example echolocation pulse where to show most of acoustic variables measured in this study. As follow from Tables S1 and S2, the authors regularly measured the peak frequency instead of the fundamental frequency, and often measured the frequency of the low-frequency noise instead of the peak frequency and bandwidth frequencies. The authors must re-measure each echolocation call manually or automatically with full visual control by spectrogram, after preliminary filtering the low-frequency noise. As the measurements of only 320 echolocation pulses are included in the manuscript, the manual re-measuring of their acoustic parameters will only take approximately 3 working days. After re-measuring, all statistics for comparison the acoustic parameters of the pulses between sexes and individuals must be re-calculated, with inclusion the actual parameter values in the analyses.
Detailed comments
L 74-75 Their echolocation calls predominantly consist of a long constant frequency (CF) component, which is often initiated and terminated by brief frequency-modulated (FM) sweeps of a substantial bandwidth.
Please provide a reference to the paper displaying spectrogram of this call or add a figure with the spectrogram in the current manuscript.
L 99-100 Thus, eight (four males and four females) and 20 adult greater horseshoe bats (10 males and 10 females) were collected in Dalazi Cave and Xin Cave, respectively.
I did not understand this sentence until reading methods below. Please correct this sentence as “Thus, eight adult greater horseshoe bats (four males and four females) were collected in Dalazi Cave and 20 adult greater horseshoe bats (10 males and 10 females) were collected in Xin Cave
L 123 Ultrosoundgate
Typo, replace with UltraSoundGate
L 126 notebook computer
Regarding the experimental setup, the authors should indicate whether they used for their recording and playback equipment the autonomous feeding (accuses or batteries) or room electric power grid. This is critically important because electric power unit per se can produce very intense ultrasonic noises, affecting bat behaviour.
L 128 at least five sound files
Replace with At least five 60-s sound files
L 130 Editing of Sex Recognition Playback Files
Consider rewriting to “Playback Stimuli for Sex Recognition”
L 136-137 and 146-147 to obtain playback sequences that were 35 s long, i.e., 15 s of echolocation calls + 5 s of silent interval + 15 s of echolocation calls
Creating the playback sequencies is poorly described and remains unclear for the reader. Above, you write that, from each individual, you selected five series of pulse trains, each 50 ms long. So, even if all series from all the 8 individuals were taken, their total duration would be 50 ms * 5 * 8 = 2000 ms = 2 s. Please describe in detail, how you created the 15-s sequences for the playbacks. Please indicate how many pulse trains from how many individual bats were included in them. Please indicate whether you used the same pulse train repeatedly within the same playback sequence. Please indicate whether the first playback sequence differed from the second one.
L 138-139 Four types of playback files were assembled for the sex recognition experiment: ♂♂, ♂♀, ♀♂, and ♀♀
The authors should obligatory provide the figure with spectrogram of example playback stimulus for sex recognition, ideally with supplementary ultrasonic audio file(s). The same for individual recognition.
L 140 Editing of Individual Recognition Playback Files
Consider rewriting to “Playback Stimuli for Individual Recognition”
L 141-142 The recorded echolocation calls of the eight bats from the Ji’an area and the 20 bats from the Panshi area
Earlier, you indicated that the 8 bats were collected in Dalazi Cave and 20 bats were collected in Xin Cave. Please keep this terminology throughout the text. This is much easier for the reader. The authors know that these two caves correspond to Ji’an area and Panshi, but the readers not. Double names for the same places are confusing for the readers, please avoid synonyms.
L 144 that had a high signal-to-noise ratio (>30 dB)
Was this criterion of intensity also applied to playback stimuli for sex recognition? Please indicate this.
L 145 assembled them in accordance with natural intervals
What are natural intervals for this species? Please provide references.
L 147-148 Five different playback files were edited for each individual for subsequent analysis and playback
What do you mean under subsequent analysis? Acoustic analysis? For what purpose?
L 153-154 Prior to conducting the acoustic analysis, the pulses were standardized to 75% to evaluate the quality of the waveforms and exclude overloaded signals [35].
I looked over the cited paper (Sun et al., 2018). This paper does not contain the description of the mentioned procedure of pulse standardization to 75%. I did not ever hear about this method. Please provide information about the conducted procedure in the current manuscript.
L155-156 Measurements were taken of spectrograms generated using a 1024-point fast Fourier transformation and a Hamming window with 75% overlap
Please also indicate frame size.
L 157-158 the duration (s) and root mean square (no unit) of the harmonics at the energy maximum were measured
The duration of the harmonics sounds strange. Do you mean call duration? If yes, please add “call” before “duration”. “root mean square (no unit) of the harmonics” should be decoded. What does this parameter mean from the bioacoustical viewpoint?
L 157-161. To quantify the sex characteristics of the echolocation calls, the duration (s) and root mean square (no unit) of the harmonics at the energy maximum were measured, as well as five spectrum-based parameters (kHz): the peak frequency, fundamental, minimum frequency, maximum frequency, and bandwidth.
Please indicate, where and how these acoustic parameters were measured. Did you apply Avisoft using Automatic Parameter Measurements? Root mean square is missing in the Automatic Parameter Measurements option of Avisoft (I have version 5.3.00). Minimum frequency – rename to minimum frequency of bandwidth; maximum frequency – rename to maximum frequency of bandwidth to make these terms clear for the readers unfamiliar with Avisoft. Please indicate the threshold for measuring bandwidth (10 dB, 20 dB, other.) All measurements of fundamental (fundamental frequency) must be deleted from analysis, because the Automatic Parameter Measurements of Avisoft measures this parameter very inaccurately and it can only be used under full visual control by researcher (see details below).
In the Methods, it is necessary to provide a figure with spectrogram of the echolocation call of the greater horseshoe bat and to indicate on it the points of measurements and the measured acoustic parameters.
L 157-161. To quantify the sex characteristics of the echolocation calls, the duration (s) and root mean square (no unit) of the harmonics at the energy maximum were measured, as well as five spectrum-based parameters (kHz): the peak frequency, fundamental, minimum frequency, maximum frequency, and bandwidth. These parameters are important and have been commonly used to determine whether acoustic signals have an individual signature [33, 36, 37].
These two sentences are poorly concordant to each other. In the first one, you say that you quantify the sex characteristics, whereas in the second one, you say that these parameters are commonly used to determine whether acoustic signals have an individual signature. I suggest to delete the second sentence from Methods and to transfer it to Discussion.
L 157-161. To quantify the sex characteristics of the echolocation calls, the duration (s) and root mean square (no unit) of the harmonics at the energy maximum were measured, as well as five spectrum-based parameters (kHz): the peak frequency, fundamental, minimum frequency, maximum frequency, and bandwidth. These parameters are important and have been commonly used to determine whether acoustic signals have an individual signature [33, 36, 37]. For these five spectrum-based parameters, three locations (i.e., the center, end, and maximum amplitude of the element) were measured [38-40]. Thus, a total of 17 acoustic variables were extracted: the duration, root mean square (rms), peak frequency (end), fundamental (end), minimum frequency (end), maximum frequency (end), bandwidth (end), peak frequency (center), fundamental (center), minimum frequency (center), maximum frequency (center), bandwidth (center), peak frequency (maximum), fundamental (maximum), maximum frequency (maximum), minimum frequency (maximum), and bandwidth (maximum).
Choice of the set of the acoustic parameters is very unsuccessful. Echolocation calls of bats are different in the acoustic structure. In the horseshoe bats, the echolocation calls are constant-frequency (CF) pulses, in which the maximum of energy goes on the second frequency band = f1 (named as the first harmonic in the Europe and the second harmonic in America). From the 17 acoustic parameters measured in this manuscript, 15 are related to the frequency of f1, whereas the duration of different call parts was not measured and was not included in analysis. The authors must substantiate the selected set of acoustic parameters (aside the argument regarding the simplicity of the automatic measurements) without referencing to the sets of parameters used for measuring the echolocation calls of different structure from other bat species.
L 175 were conducted in an experimental cage
Please indicate material of which the cage was made.
L 188-189 One type of playback file (5 ♀ files or 5 ♂ files) was selected randomly
Previously (L 138-139) you wrote that Four types of playback files were assembled for the sex recognition experiment: ♂♂, ♂♀, ♀♂, and ♀♀. Please make clear what files were prepared and used for playbacks.
L 240 calls for individual recognition, six principal components
Please adjust that above you described the principal components for sexual recognition.
L 281 all of the female echolocation calls
Replace "all" with "the average values"
L 282-283 see Supplementary Material, Table S1
The main problem of this manuscript is in mistaken measuring the Fundamental frequency using the Automatic Parameter Measurements in Avisoft. In the horseshoe bats, the fundamental frequency of the echolocation pulses is always twice lower than the peak frequency, and does not exceed 50 kHz (see spectrograms in Jones & Rayner 1989; Tian & Schnitzler 1997; Ma et al. 2006; Bastian & Jacobs 2015; Finger et al. 2017). So, the provided by the authors average values for the fundamental (center) and fundamental (maximum) (Table S1) indicate that, in many cases, the authors automatically measured the peak frequency instead of the fundamental frequency. This is an uncorrectable mistake of the Automatic Parameter Measurements for the duration of 20 years, so please avoid using the automatic measuring the fundamental frequency with Avisoft.
L 317-317 see Supplementary Material, Table S2
Table S2 also indicated the cruel mistakes of the automatic measurements of the acoustic parameters using the Automatic Parameter Measurements. The minimal values of the peak frequency cannot be lower than 50 kHz. The minimal values of the fundamental frequency cannot be lower than 25 kHz (see spectrograms). The minimal and maximal values of bandwidth frequency must be a bit lower or a bit higher the values of the peak frequency. The values of 2, 3, 7, 11 kHz indicate that the authors measured the frequency of the background noise instead of the peak or fundamental frequencies.
The authors must re-measure the echolocation calls, after the preliminary filtering out the low-frequency noise with high-pass filter). When measuring the peak frequency and bandwidth frequencies, is it necessary to turn on the visualization (Additional Spectrogram Information - Spectrogram - Maximum) and to visually control the peak frequency in each echolocation pulse, especially in position "end", where call intensity is weak. Otherwise, you can make the manual measurements of the peak frequency and the bandwidth frequencies, at which you will immediately see the mistakes of the measurements and can correct them. Regarding the fundamental frequency, all measurements should either be deleted from analysis or replaced with manual measurements in the selected points of spectrogram. You measured 320 echolocation pulses, this is approximately 3 days of work at the manual measurements of all acoustic parameters.
L 315-316 which is significantly higher than the random classification probability (10.0%, binominal test: P < 0.01)
The random value here is not 10%. In your analysis, nonindependent data are included (multiple measurements of the peak frequency). Correct calculations of random value see in
Solow A.R., 1990. A randomization test for misclassification probability in discriminant analysis. Ecology, 71:2379–2382.
Mundry R., Sommer C., 2007. Discriminant function analysis with nonindependent data: consequences and an alternative. Animal Behaviour, 74:965-976.
L 327-328 compared to transforming different acoustic files for the same individual
What do you mean? Please explain what the transforming files mean
L 345-348 Several studies on birds (e.g., Alca torda [8] and Calonectris diomedea [50]) and mammals (e.g., Mus musculus musculus [51]) have also shown that their acoustic signals can contain sex information, and thus, other individuals can recognize the sex of the conspecifics by their calls.
For birds, see the relevant review of sex differences in the acoustic signals in Volodin et al., 2015. Gender identification using acoustic analysis in birds without external sexual dimorphism, Avian Research, 6:20. The reference [51] (Zala et al., 2020) is irrelevant here, because it is only on male mice ultrasonic vocalizations.
L 369-370 Low DFA classification rates for individual calls have also been reported in other studies [62, 63].
Replace “studies” with “mammals”. DFA classification rates for individual calls in birds can be very high.
L 371-372 Most mammals encode their acoustic individual characteristics with source characteristics such as frequency, amplitude contours, and harmonic structure [63].
Most mammals? The study [63] is on a single species, the giant otter. This is confusing; please add a more relevant reference here, preferentially a couple of recent reviews.
L 373-375 In this study, we only measured some of the parameters of the harmonic containing the most energy in the call, which cannot realistically represent the entire acoustic differences among the individuals.
This sentence is unclear. What do you mean?
L Supplementary Materials
Insert the supplementary Tables to the main body of the manuscript.
Reviewer 3 Report
I think this was a clear, accessible study that did a good job tying into past research on the topic. The use of references was very good and seemed to have a good coverage of the field. The explanation of the experiments was clear and did not seem to have any problems with clarity. There were some minor issues with citations. For example, on likes 65 and 66, the claims is made that reference 17 provides evidence that bats can recognize species identity, However, this reference was only looking at techniques for humans to assess bats (using DFA and neural networks) but made no attempt to see if bats could discriminate species. Similarly, only line 67, reference 25 is listed as providing evidence for individual identification, when it was for sex, not individual (so the reference should be moved to earlier in the sentence). There was also a reference that I think should be included - it used a similar playback technique as the current study to determine if big brown bats could determine sex based solely on echolocation calls. The paper is Kazial and Masters (2004) doi:10.1016/j.anbehav.2003.04.016. While the species was different, the technique was so similar it seems that addressing that study would be worthwhile.
Round 2
Reviewer 2 Report
The authors addressed all my comments and thoroughly improved the manuscript. I found however a few minor mistakes which should be obligatory corrected before publishing.
L 317 Table 1
Among the measured acoustic parameters you twice indicate Peak frequency (end) and Minimum frequency of bandwidth (end) but do not indicate Maximum frequency of bandwidth (maximum) and Bandwidth (maximum). Please correct.
L 537 Make in italic Anim. Behav. and 74
L 552 Replace “Ilya, V.; Elena V.; Anna K.; Vera M.,” with “Volodin, I.A.; Volodina, E.V.; Klenova, A.V.; Matrosova, V.A.,”
Replace 6: with 6, (in italic)
L 581
Replace 9: with 9, (in italic)
Author Response
L 317 Table 1 Among the measured acoustic parameters you twice indicate Peak frequency (end) and Minimum frequency of bandwidth (end) but do not indicate Maximum frequency of bandwidth (maximum) and Bandwidth (maximum). Please correct.
Response: Thank you for finding the problem. We have corrected the Table 1. Please see Table 1 in the revised manuscript.
L 537 Make in italic Anim. Behav. and 74
Response: Thank you very much. We made in italic Anim. Behav. and 74. Please see line 537.
L 552 Replace “Ilya, V.; Elena V.; Anna K.; Vera M.,” with “Volodin, I.A.; Volodina, E.V.; Klenova, A.V.; Matrosova, V.A.,”
Replace 6: with 6, (in italic)
Response: We replaced “Ilya, V.; Elena V.; Anna K.; Vera M.,” with “Volodin, I.A.; Volodina, E.V.; Klenova, A.V.; Matrosova, V.A.,” and replaced 6: with 6, (in italic). See line 552.
L 581 Replace 9: with 9, (in italic)
Response: We did it. Please see line 581.